# The Sustainability of Corporate ESG Performance: An Empirical Study

Kezhi Yang [1,*], Tingting Zhang [1] and Chenyun Ye [2]

1. School of Business, Beijing Technology and Business University, Beijing 100048, China; 17690921072@163.com
2. Accounting School, Shandong Management University, Jinan 250100, China; yechenyun7290@163.com
* Correspondence: yangkezhi@btbu.edu.cn; Tel.: +86-152-1088-8684

**Abstract:** A company's ESG (environmental, social, and government) performance is an indicator of its sustainable development. In practice, enterprises should focus on improving their governance structure and improving their governance level to achieve sustainable development and long-term value. Based on a sample of China's A-share-listed companies from 2014 to 2022, this paper obtains data from the WIND and CSMAR databases and finally selects 14,757 observed values. With ESG performance as the explained variable and Pledge as the explanatory variable, the relationship between major shareholders' equity pledges and ESG performance is explored using a regression analysis. The results show that the correlation coefficient, $\beta 1$, between corporate ESG performance and the pledge ratio of major shareholders is $-0.0167$, which is significantly negative at the 1% level, indicating that the equity pledges of major shareholders will have a negative impact on corporate ESG performance, and ESG performance shows that the pressure of controlling shareholders' equity pledges mainly reduces the performance of companies in the areas of social responsibility (S) and governance (G) and does not have a significant impact on environmental construction (E). Further research shows that under the same conditions, compared with state-owned enterprises, the equity pledge behavior of major shareholders of private enterprises has a more significant impact on corporate ESG performance. This study is a good attempt at examining the sustainability of corporate ESG performance.

**Keywords:** ESG (environment, society, and government); sustainable development; equity pledging





## 1. Introduction

In recent years, as the global population has continued to grow and the aging of the population has accelerated, we have faced many problems such as global warming, the COVID-19 pandemic, and international conflicts. The importance of corporate sustainable development is increasing day by day, and ESG performance, which comprehensively considers the three dimensions of environmental (E), social (S) and governance (G) performance to evaluate corporate sustainable development, has become an important investment concept and corporate action guide around the world. Through ESG performance, enterprises can effectively evaluate their contributions to fulfilling their social responsibilities and promoting green and sustainable development [1]. The concept of ESG performance is gradually becoming a trend in enterprise development. The large scale of Chinese companies, the behavior characteristics of consumers, which have a significant impact on market demand, increasingly fierce market competition, and the policy and regulatory environment of the Chinese market have a significant impact on market development. On this basis, ESG research in this market is more attractive. Improving enterprise ESG performance has become an unavoidable issue for sustainable development.

An equity pledge is a type of commitment behavior established by the shareholders of listed companies in exchange for funds from banks or other financial institutions. Its significant feature is that shareholders still have voting rights and control rights while

receiving funds [2,3]. Compared with a traditional mortgage, an equity pledge has the advantages of fewer approval procedures, that the right of control will not be diluted, and so on; it has produced explosive growth in recent years. Anderson and Puleo [4] conducted a random selection of 500 companies from the S&P 1500 index in the United States, revealing that approximately 16% of these companies had insiders (senior management and directors) pledging equity. Similarly, some scholars discovered that around 40% of shareholders in listed Indian companies had pledged equity. Because a major shareholder plays an important controlling role in an enterprise, their equity pledge behavior will have a certain impact on the enterprise. If the stock price falls to the level line after the major shareholder's equity pledge, the shareholder cannot add the margin in a timely manner; they will face the risk of control being transferred. In order to avoid the transfer of control, major shareholders have the ability to put pressure on management, intervening by manipulating earnings, reducing the disclosure of negative information, and using other measures [5,6] to prevent the share price from falling more than expected. Enterprise ESG performance is a high-investment, long-cycle project that cannot bring high returns to the enterprise in a short period of time, which is not conducive to the short-term performance of the enterprise. So, will the equity pledges of major shareholders inhibit the ESG performance of enterprises? Solving the above problems has high research value and practical significance for promoting the green development of global enterprises.

Equity pledging by major shareholders and corporate environmental, social, and governance (ESG) performance have gained significant public attention. The concept of ESG performance is relatively nascent, having evolved from the earlier notion of shareholder supremacy, which advocated that companies should prioritize shareholder interests and solely focus on maximizing profits or shareholder value [7]. However, as dissatisfaction with shareholder supremacy grew, stakeholder theory gained prominence, positing that companies are comprise various stakeholders whose interests should be considered alongside those of shareholders [8]. Companies could be accountable not only to major shareholders but to other stakeholders such as suppliers and consumers. In response, the business community has shifted its stance and now emphasizes support for corporate social responsibility [9,10]. Against this backdrop, exploring the relationship between major shareholders' equity pledging and corporate ESG performance has both theoretical and practical significance for industry practices and norms.

This paper takes A-share-listed companies from 2014 to 2022; using a regression analysis method, it explores the impact of the equity pledging of major shareholders on corporate ESG performance and its channels of action and considers what changes will occur in this impact under different scenarios. This paper's main contributions are reflected in the three aspects listed below.

First, this study contributes to research on the economic implications of equity pledging. While numerous existing studies have investigated the effects of major shareholders' equity pledging on corporate financial health, innovation, and overall value, this paper redirects attention toward corporate environmental, social, and governance (ESG) performance. By examining the influence of equity pledging on corporate decision-making through the lens of ESG performance, this study offers a more comprehensive understanding and provides additional evidence to the existing body of literature.

In addition, this study contributes to the determinants of corporate environmental, social, and governance (ESG) performance. The existing literature examines the economic implications of corporate ESG performance, with few studies examining influencing factors. Moreover, those studies often adopt an external perspective, emphasizing environmental and policy effects. In contrast, this paper takes an approach of investigating the impact of major shareholders' equity pledging on corporate governance, highlighting its dual nature and how it affects corporate ESG performance from the perspective of shareholder behavior.

Furthermore, this study delineates the causal pathway linking major shareholders' equity pledging to corporate environmental, social, and governance (ESG) performance.

By examining the influence of both internal and external oversight mechanisms and considering variations in impact across different property rights structures, the insights have theoretical significance for advancing corporate governance effectiveness and practical implications for enhancing ESG performance within organizations.

## 2. Theoretical Framework

### 2.1. Research on the Economic Consequences of Major Shareholders' Equity

As a type of financing method with a low cost and high efficiency, equity pledging plays an important role in resolving the financing problems of shareholders and shortages of funds in enterprises [11] and has become an increasingly favored financing means for enterprises. Existing studies have revealed that the role of the equity pledges of major shareholders in corporate governance has two sides [12,13].

On one hand, an equity pledge will produce a certain negative effect on behavior. In terms of innovation investment, Lu et al. [14] argued that while equity pledging is intended to mitigate financial constraints, it fails to incentivize enterprises to increase investments in innovation. This reluctance to innovate may stem from major shareholders' concerns about losing control of the enterprise in the event of failed innovation endeavors [15]. Concerning corporate value, major shareholders pledging equity exacerbates agency problems and diminishes the market value of listed companies [16,17]. Su et al. further demonstrated that a greater divergence between the control rights and cash flow rights of major shareholders correlates with a decline in corporate value. Moreover, a higher proportion of equity pledged results in a higher sensitivity coefficient for encroaching corporate value interests [18].

On the other hand, an equity pledge will produce certain positive effects. In terms of corporate performance, Wang and Chou [17] argued that equity pledging can enhance a company's investment opportunities, thus contributing to improved corporate performance. Regarding stock prices, some experts suggested that equity pledging prompts listed companies to capitalize development expenditures, thereby conveying positive signals to the capital market [19] which may bolster or sustain the company's stock price. Regarding stock price crash risk, the risk stock price crash risk is lower while there are equity pledges and increases after the pledges are lifted. Regarding accounting policies, Lu and Williams [20] argued that after equity pledging, listed companies tend to convey "good news" to the market through more aggressive accounting policies and other means. Concerning corporate value, Li et al. [21] found that major shareholders pledging equity enhances company value through an increased convergence of interests and the signal transmission effect of major shareholders' optimistic stance toward stock prices.

### 2.2. Research on the Factors Influencing Enterprise ESG Performance

Numerous existing studies have investigated the factors influencing corporate environmental, social, and governance (ESG) practices, primarily examining the internal and external factors surrounding companies.

From the perspective of a company's internal environment, Zhou et al. [22] and DasGupta [23] asserted that companies with exemplary performance demonstrate superior environmental, social, and governance (ESG) practices. They argue that exceptional performance serves as an economic foundation for fulfilling social and environmental responsibilities. Focusing on Malaysia and drawing from stakeholder theory and agency theory [24], they discovered that ESG certification reduces a company's capital costs, enhances its Tobin Q value, and elevates its market value. Taking Italian food companies as a research object, it was found that sustainability in sustainability reports is related to corporate value [25]. Similarly, Pozzoli et al. [26] and Rego and Wilson [27] found that an audit committee can play an internal supervision role, improve the transparency and accuracy of a company's financial report, and significantly improve the company's ESG performance. Liu and Zhang [28] showed that the short-sighted authorities of a company's

management are prone to opportunistic behaviors which will have a negative impact on the company's ESG performance.

From the perspective of a company's external environment, the industry sector and financial variables serve to identify significant differences across firms regarding ESG performance [29]. Chan et al. [30] found that the development of the urban digital economy can improve innovation output capacity and improve a company's ESG performance. Examining ESG ratings, Serafeim and Yoon [31] utilized the ESG rating index by MSCI (Morgan Stanley Capital International) to investigate the correlation between stock prices and ESG information. Investors and the other financial stakeholders remain the key stakeholders of many organizations, and they continue to represent the primary recipients of corporate reports [32,33]. They found that consensus ESG ratings can forecast future market trends, enabling corporate investors to adjust their investment strategies promptly and consequently enhance corporate financial performance.

The existing literature contains a detailed study on the economic effects of equity pledging by major shareholders, revealing its two-sided impact on corporate governance, but it has not paid much attention to the topic of how equity pledging affects performance with respect to corporate ESG responsibility. However, in the field of enterprise ESG research, scholars pay more attention to the positive effects on enterprises of good ESG performance and reveal its positive role in enterprise financing, innovation, and performance but ignore the exploration of factors affecting performance with respect to enterprise ESG responsibility. Only a few relevant studies in the literature mainly focus on the external environments or institutional backgrounds of enterprises. Little attention has been paid to the role of internal factors in a company's ESG performance. Therefore, this paper attempts to shift the research perspective to the level of corporate governance and explore the impact of the equity pledges of major shareholders on performance with respect to corporate ESG responsibility and its mechanism.

### 2.3. Literature Review and Hypotheses Development

Major shareholders' equity pledges may adversely affect the ESG performance of enterprises in the following two ways [34]. First, the equity pledges of major shareholders will reduce the willingness of enterprises to fulfill ESG responsibilities. According to the principal–agent theory, when the equity of an enterprise is concentrated and the cash flow right enjoyed by the controlling shareholder is smaller than its control right, in order to obtain economic benefits [35], the major shareholder tends to seize the interests of minority shareholders and "hollow out" the enterprise through "tunnel digging" such as related party transactions and the non-public offering of shares [36]. After an equity pledge, non-property rights such as control rights are still enjoyed by the controlling shareholder, but the cash flow rights corresponding to the pledged shares are transferred to the pledgee, which further aggravates the degree of separation between the two rights and further aggravates the motivation to hollow out [37]. Opportunistic behaviors of major shareholders such as hollowing out are short-sighted behaviors [38] which not only increase the occupation of corporate resources but also cause them to tend to invest in projects with high self-interest and low risk [39] in investment decisions, thus weakening their willingness to bear long-term risks. As a long-term strategic decision, performing ESG responsibilities has slow returns and a high degree of uncertainty, and the short-sighted tendency makes it difficult for shareholders to foresee the long-term positive effects of performing ESG responsibilities on corporate value; therefore, they will reduce their investment in such projects contrary to their objectives, thus inhibiting improvement in the level of performing ESG responsibilities. Second, the equity pledges of major shareholders will reduce the ability of enterprises to fulfill their ESG responsibilities. An equity pledge itself means that the controlling shareholder is facing financial difficulties. Although financing through an equity pledge improves the controlling shareholder's own capital shortage, the cash flow of the enterprise is not increased significantly because the funds obtained are mostly used by the shareholders themselves or third parties [40]. In addition,

after the occurrence of an equity pledge, in order to prevent a loss of control caused by falling stock prices, the opportunistic behaviors of controlling shareholders based on market value management motivation are strengthened, mainly manifesting in the use of earnings manipulation, selective information disclosure, etc. Such potential opportunistic behaviors aggravate the information asymmetry between enterprises and external fund suppliers. Faced with potential moral hazards of enterprises, capital suppliers will demand higher risk premiums, resulting in more serious financing constraints for enterprises. According to the resource supply hypothesis, enterprises need to have sufficient financial strength to undertake activities such as social responsibility activities. Therefore, when an equity pledge cannot effectively alleviate the financial pressure of enterprises but worsens their financing difficulties, their ESG performance level may be inhibited. Building upon the aforementioned analysis, this paper posits Hypothesis 1:

**H1.** *The equity pledging of major shareholders has a negative impact on corporate ESG performance.*

The analysis above elucidates several key points.

Firstly, following major shareholders' equity pledging, the pressure stemming from such pledging may lead to their increased intervention in the company's daily operations and management, resulting in a scenario in which a single shareholder dominates decision making. This dominance infringes upon the interests of small and medium shareholders, exacerbates the company's agency problems, and diminishes the quality of internal control, consequently affecting the company's governance standards.

Secondly, as pressure from major shareholders' equity pledging escalates, the company's available funds for social responsibility investments diminish, while the information asymmetry resulting from concealing adverse news deteriorates the company's external financing environment. Additionally, engaging in earnings manipulation, which is deemed improper behavior, further erodes accountability to external investors and society, thereby reducing the company's level of social responsibility performance.

Thirdly, owing to the prolonged payback period and heightened risk associated with green technology innovation investments, major shareholders, in a bid to retain control rights, are inclined to opt for low-risk, short-term investments, consequently stifling the company's allocation of resources toward green technology innovation.

From this analysis, it is evident that major shareholder equity pledging leads to reductions in a company's governance level, social responsibility performance, and investments in environmental initiatives, consequently impacting the company's environmental, social, and governance (ESG) performance. Building upon this analysis, the following hypotheses are proposed:

**H1a.** *The equity pledging of major shareholders will reduce a company's environmental performance (E).*

**H1b.** *The equity pledging of major shareholders will reduce a company's social performance (S).*

**H1c.** *The equity pledging of major shareholders will reduce a company's governance level (G).*

## 3. Methodology

### 3.1. Sample and Data Source

This paper utilizes annual data from 2014 to 2022 of A-share-listed companies in China to investigate the relationship between major shareholder equity pledging and corporate environmental, social, and governance (ESG) performance. ESG-related data are sourced from the WIND database, while other financial data primarily originate from the CSMAR database. The original data undergo preliminary screening through the following steps: (1) the exclusion of samples from the financial and real estate industries; (2) the exclusion of samples from ST- and PT-listed companies; (3) the elimination of samples with missing

variables; and (4) the use of Winsorization tail truncation on continuous data beyond 1% to mitigate the impact of outliers. The purpose of data screening is to improve the availability of relevant data collected and stored before, which is more conducive to later data analyses. Following screening, a total of 14,757 observations are ultimately retained for analysis.

*3.2. Research Model*

3.2.1. Model Setting

To test the impact of major shareholders' equity pledging on corporate ESG performance (H1), we construct model (1) as follows:

$$
\begin{aligned}
\text{ESG}_{i,t} &= \alpha_0 + \alpha_1 \text{Pledge}_{i,t} + \alpha \text{Control}_{i,t} + \sum \text{Year} + \sum \text{Ind} + \varepsilon_{i,t} \\
\text{E}_{i,t} &= \alpha_0 + \alpha_1 \text{Pledge}_{i,t} + \alpha \text{Control}_{i,t} + \sum \text{Year} + \sum \text{Ind} + \varepsilon_{i,t} \\
\text{S}_{i,t} &= \alpha_0 + \alpha_1 \text{Pledge}_{i,t} + \alpha \text{Control}_{i,t} + \sum \text{Year} + \sum \text{Ind} + \varepsilon_{i,t} \\
\text{G}_{i,t} &= \alpha_0 + \alpha_1 \text{Pledge}_{i,t} + \alpha \text{Control}_{i,t} + \sum \text{Year} + \sum \text{Ind} + \varepsilon_{i,t}
\end{aligned}
\tag{1}
$$

3.2.2. Variable Definition

1.   Explained variable

Corporate environmental, social, and governance (ESG) performance (ESG) is assessed using the Huazheng ESG evaluation score. The Huazheng ESG evaluation consists of nine levels: AAA, AA, A, BBB, BB, B, CCC, CC, and C, with corresponding values ranging from 9 to 1 to represent corporate ESG performance. A higher score indicates better corporate ESG performance. Environmental governance performance (E) is determined by averaging the quarterly ratings of the Huazheng ESG score specifically pertaining to environmental factors. Social responsibility performance (S) and corporate governance performance (G) are evaluated using the same criteria as environmental governance performance (E).

2.   Explanatory variables

Following the methodology outlined by Hu Jun et al., the equity pledge ratio (Pledge) is employed to quantify the extent of equity pledging within the sample companies.

3.   Control variables

Drawing on prior research [41,42], this study controls for the following variables that could potentially influence corporate environmental, social, and governance (ESG) performance: company size (Size), company age (Age), leverage ratio (Lev), cash flow (Cfo), growth (Growth), the shareholding of the largest shareholder (Top1), board independence (Rind), and board size (Board). Detailed descriptions of these variables are provided in Table 1.

**Table 1.** Variable declaration.

| Variable | Variable Name | Variable Symbol | Variable Description |
|---|---|---|---|
| Explained variable | Corporate ESG rating | ESG | According to China Securities 9 file rating, ratings from high to low are assigned values of 9~1 |
| | Environmental rating | E | The average of ESG's quarterly environment (E) ratings |
| | Social score | S | The average of ESG's quarterly social (S) ratings |
| | Corporate governance score | G | The average of ESG's quarterly corporate governance (G) ratings |
| Explanatory variable | Share pledge ratio | Pledge | Number of shares pledged by major shareholders/total number of shares held |

**Table 1.** *Cont.*

| Variable | Variable Name | Variable Symbol | Variable Description |
|---|---|---|---|
| Control variable | Enterprise scale | Size | The natural log of total assets at the end of the period |
| | Enterprise age | Age | The logarithm of the number of years the company has been listed plus one |
| | Leverage ratio | Lev | Total liabilities/total assets |
| | Cash flow | Cfo | Cash flow from operating activities divided by total assets |
| | Growth | Growth | Growth rate of main business income |
| | The largest shareholder holds shares | Top1 | The proportion of the largest shareholder |
| | Board independence | Rind | Number of independent directors/board size |
| | Board size | Board | Number of directors |
| | Individual fixation effect | Ind | Industry dummy variables; refer to China Securities Regulatory Commission 2012 Industry classification standard |
| | Annual fixed effect | Year | Annual dummy variable |

## 4. Results and Discussion

### 4.1. Descriptive Statistics

Table 2 presents the descriptive statistical findings of this study. The dataset comprises 14,757 observations. Upon analyzing all observed values, it is revealed that the mean and median of environmental, social, and governance (ESG) performance (ESG) for A-share listed companies during the period of 2014–2022 are 3.944 and 4.000, respectively, with a standard deviation of 1.109. This indicates that while most companies exhibit satisfactory ESG performance, there remains significant room for improvement. The minimum value of 1.000 suggests that certain enterprises demonstrate poor ESG performance and necessitate further enhancement. Moreover, the average values of environmental governance performance (E), social responsibility performance (S), and corporate governance performance (G) are 1.916, 4.028, and 4.957, respectively; this shows that the companies' performance at the levels of social responsibility and governance is better, while their performance at the environmental level is poor, which can also reflect that Chinese companies might pay attention to the environment last and investment in the environment has just begun. Furthermore, the average value of the equity pledge ratio (Pledge) is 14.564, highlighting the prevalence of major shareholder equity pledging among listed companies. The observed values of other control variables align with the existing literature and exhibit relative stability.

**Table 2.** Descriptive statistics of variables.

| VarName | Obs | Mean | SD | Min | Median | Max |
|---|---|---|---|---|---|---|
| ESG | 14,757 | 3.944 | 1.109 | 1.000 | 4.000 | 6.000 |
| E | 14,757 | 1.916 | 1.154 | 1.000 | 1.000 | 6.000 |
| S | 14,757 | 4.028 | 1.150 | 1.000 | 4.000 | 7.000 |
| G | 14,757 | 4.957 | 1.473 | 1.000 | 5.000 | 8.000 |
| Pledge | 14,757 | 14.564 | 14.340 | 0.000 | 10.790 | 63.060 |
| Size | 14,757 | 8.159 | 1.125 | 5.915 | 8.024 | 11.858 |
| Age | 14,757 | 7.610 | 0.001 | 7.608 | 7.610 | 7.612 |
| Cfo | 14,757 | 0.046 | 0.067 | −0.187 | 0.045 | 0.284 |
| Lev | 14,757 | 0.393 | 0.194 | 0.037 | 0.381 | 0.900 |
| Growth | 14,757 | −0.191 | 0.353 | −0.927 | −0.199 | 1.877 |
| Top1 | 14,757 | 30.233 | 13.233 | 6.870 | 28.460 | 71.440 |
| Rind | 14,757 | 0.384 | 0.066 | 0.250 | 0.375 | 0.600 |
| Board | 14,757 | 9.089 | 2.264 | 5.000 | 9.000 | 18.000 |

### 4.2. Correlation Analysis

Table 3 presents the correlation analysis outcomes for the selected variables. The correlation analysis reveals that the correlation coefficient (β1) between corporate environmental, social, and governance (ESG) performance (ESG) and the equity pledge ratio of major shareholders (Pledge) is −0.255, exhibiting a significant negative correlation at the 1% level. This finding suggests that the equity pledge ratio of major shareholders may contribute to a decline in corporate ESG performance. A possible reason for this is that an equity pledge worsens the controlling shareholder's motivation to hollow out and aggravates the financing dilemma of enterprises so enterprises will reduce their ESG investments, resulting in poor performance with respect to their ESG responsibilities. Consequently, this supports research hypothesis H1 posited in this paper, suggesting that the equity pledging of major shareholders has an adverse effect on corporate ESG performance. However, further empirical testing is required to ascertain the specific nature of this relationship.

**Table 3.** Correlation analysis results.

| VarName | ESG | E | S | G | Pledge | Size | Age | Cfo | Lev | Growth | Top1 | Rind |
|---|---|---|---|---|---|---|---|---|---|---|---|---|
| ESG | 1.000 | | | | | | | | | | | |
| E | 0.482 *** | 1.000 | | | | | | | | | | |
| S | 0.947 *** | 0.456 *** | 1.000 | | | | | | | | | |
| G | 0.663 *** | 0.049 *** | 0.613 *** | 1.000 | | | | | | | | |
| Pledge | −0.255 *** | −0.001 | −0.249 *** | −0.369 *** | 1.000 | | | | | | | |
| Size | 0.121 *** | 0.203 *** | 0.136 *** | −0.017 ** | −0.006 | 1.000 | | | | | | |
| Age | 0.007 | 0.097 *** | 0.120 *** | −0.167 *** | 0.039 *** | 0.158 *** | 1.000 | | | | | |
| Cfo | 0.133 *** | 0.040 *** | 0.114 *** | 0.170 *** | −0.108 *** | 0.048 *** | 0.045 *** | 1.000 | | | | |
| Lev | −0.137 *** | 0.104 *** | −0.097 *** | −0.283 *** | 0.110 *** | 0.499 *** | 0.082 *** | −0.175 *** | 1.000 | | | |
| Growth | −0.036 *** | −0.052 *** | −0.025 *** | −0.001 | 0.013 | 0.085 *** | −0.002 | 0.073 *** | 0.001 | 1.000 | | |
| Top1 | 0.058 *** | −0.041 *** | 0.035 *** | 0.140 *** | −0.029 *** | 0.066 *** | −0.153 *** | 0.096 *** | −0.003 | 0.051 *** | 1.000 | |
| Rind | 0.063 *** | −0.011 | 0.058 *** | 0.105 *** | 0.021 *** | −0.079 *** | 0.013 | 0.014 * | −0.059 *** | −0.012 | 0.027 *** | 1.000 |
| Board | −0.028 *** | 0.043 *** | −0.026 *** | −0.052 *** | −0.054 *** | 0.211 *** | −0.053 *** | 0.004 | 0.143 *** | 0.021 ** | −0.034 *** | −0.259 *** |

t statistics in parentheses; * $p < 0.1$, ** $p < 0.05$, and *** $p < 0.01$.

### 4.3. Empirical Regression Results and Analysis

Table 4 presents benchmark regression outcomes concerning the equity pledging of major shareholders and corporate environmental, social, and governance (ESG) performance. The detailed regression results are outlined below:

Columns (1), (3), (5), and (7) present the test results without controlling for the fixed effects of individual companies. The correlation coefficients (β1) between corporate environmental, social, and governance (ESG) performance (ESG), social responsibility performance (S), and corporate governance performance (G) and the equity pledge ratio of major shareholders (Pledge) are −0.0197, −0.0200, and −0.0379, respectively. These coefficients are significantly negative at the 1% level, indicating that major shareholder equity pledging may result in a decrease in corporate ESG performance, social responsibility performance (S), and corporate governance performance (G).

Furthermore, the correlation coefficient (β1) between environmental governance performance (E) and the equity pledge ratio of major shareholders (Pledge) is −0.0001, and the relationship between the two is not significant, suggesting that major shareholder equity pledging is not related to environmental governance performance (E). Columns (2), (4), (6), and (8) present the test results after controlling for the individual effects of the companies.

The correlation coefficient (β1) values between corporate environmental, social, and governance (ESG) performance (ESG), social responsibility performance (S), and corporate governance performance (G) and the equity pledge ratio of major shareholders (Pledge) are −0.0167, −0.0162, and −0.0314, respectively. These coefficients are significantly negative at the 1% level, suggesting that major shareholder equity pledging may lead to a decrease in corporate ESG performance, social responsibility performance (S), and corporate governance performance (G). Additionally, the correlation coefficient (β1) between environmental governance performance (E) and the equity pledge ratio of major shareholders (Pledge) is −0.0006, indicating no significant relationship between them.

**Table 4.** Benchmark regression results of major shareholder equity pledges and firm ESG performance.

| VarName | (1) ESG | (2) ESG | (3) E | (4) E | (5) S | (6) S | (7) G | (8) G |
|---|---|---|---|---|---|---|---|---|
| Pledge | −0.0197 *** | −0.0167 *** | −0.0001 | −0.0006 | −0.0200 *** | −0.0162 *** | −0.0379 *** | −0.0314 *** |
| | (−18.0674) | (−16.1610) | (−0.0610) | (−0.5240) | (−18.1950) | (−15.7420) | (−29.9511) | (−27.2320) |
| Size | | 0.2626 *** | | 0.2455 *** | | 0.2576 *** | | 0.1964 *** |
| | | (15.9852) | | (11.8975) | | (15.6102) | | (10.8595) |
| Age | | 46.4066 *** | | 12.9371 ** | | 46.9817 *** | | 10.6877 * |
| | | (8.8133) | | (2.0496) | | (8.8094) | | (1.7977) |
| Cfo | | 1.4140 *** | | 0.6280 *** | | 1.3561 *** | | 2.0491 *** |
| | | (8.2516) | | (3.3743) | | (7.9006) | | (10.2072) |
| Lev | | −1.2839 *** | | 0.0502 | | −1.1422 *** | | −2.2441 *** |
| | | (−14.5121) | | (0.4961) | | (−12.6704) | | (−22.4304) |
| Growth | | −0.1345 *** | | −0.1161 *** | | −0.1310 *** | | −0.0912 ** |
| | | (−3.9453) | | (−3.1811) | | (−3.8135) | | (−2.1847) |
| Top1 | | 0.0038 *** | | −0.0024 * | | 0.0035 *** | | 0.0097 *** |
| | | (3.2754) | | (−1.8267) | | (3.0376) | | (7.9675) |
| Rind | | 1.0530 *** | | 0.0864 | | 1.0028 *** | | 2.1168 *** |
| | | (5.9895) | | (0.4317) | | (5.6558) | | (10.7709) |
| Board | | −0.0162 *** | | 0.0088 | | −0.0140 *** | | −0.0269 *** |
| | | (−3.0722) | | (1.4440) | | (−2.6357) | | (−4.4908) |
| _cons | 4.2312 *** | 1.2699 *** | 1.9173 *** | −0.0305 | 4.3189 *** | 1.3392 *** | 5.5096 *** | 3.9082 *** |
| | (190.8808) | (4.4172) | (71.7221) | (−0.1268) | (193.0514) | (4.6503) | (222.1062) | (8.3042) |
| Ind | No | Yes | No | Yes | No | Yes | No | Yes |
| year | No | Yes | No | Yes | No | Yes | No | Yes |
| N | 14,757 | 14,757 | 14,757 | 14,757 | 14,757 | 14,757 | 14,757 | 14,757 |
| adj. R2 | 0.065 | 0.189 | −0.000 | 0.166 | 0.062 | 0.228 | 0.136 | 0.290 |

t statistics in parentheses; * $p < 0.1$, ** $p < 0.05$, and *** $p < 0.01$.

Based on the analysis above, the results demonstrate that the equity pledge ratio of major shareholders indeed impairs corporate ESG performance, social responsibility performance (S), and corporate governance performance (G), thus confirming hypotheses H1, H1b, and H1c, while H1a is not supported. The main reason may be that after controlling shareholders make an equity pledge, some of them will reduce the company's investment in environmental construction based on their own pledge pressure, while others tend to maintain a good image of the company through environmental construction to establish investors' confidence in the company. As a result, the influence of controlling shareholders' equity pledge pressure on a company's environmental construction is not significant. Moreover, the relationship between environmental governance performance (E) and the equity pledge ratio of major shareholders (Pledge) is found to be insignificant. Therefore, the next step involves further industry-specific analysis to explore the relationship between the equity pledge ratio of major shareholders (Pledge) and environmental governance performance (E).

As depicted in Table 5, upon industry classification, the correlation coefficient (β1) between environmental governance performance (E) and the equity pledge ratio of major shareholders (Pledge) is −0.0419, which is significantly negative at the 1% level. This confirms hypothesis H1a, indicating that the equity pledge of major shareholders diminishes a company's environmental performance (E).

**Table 5.** Benchmark regression results of major shareholder equity pledges and environmental governance performance (E).

| VarName | (1) | (2) |
|---|---|---|
| | E | E |
| Pledge | 0.0002 | −0.0419 *** |
| | (0.0372) | (−9.0364) |
| Size | | 0.2461 *** |
| | | (11.9104) |
| Age | | −0.003 * |
| | | (−0.15) |
| Cfo | | 0.6417 *** |
| | | (3.4796) |
| Lev | | 0.0426 |
| | | (0.4283) |
| Growth | | −0.1169 *** |
| | | (−3.2010) |
| Top1 | | −0.0024 * |
| | | (−1.8183) |
| Rind | | 0.0841 |
| | | (0.4200) |
| Board | | 0.0090 |
| | | (1.4660) |
| _cons | 1.9132 *** | 0.5242 * |
| | (23.3200) | (1.9157) |
| Ind | No | Yes |
| year | No | Yes |
| N | 14,757 | 14,757 |
| adj. R2 | −0.000 | 0.166 |

t statistics in parentheses; * $p < 0.1$ and *** $p < 0.01$.

### 4.4. Endogeneity and Robustness Test

To ensure the robustness of the regression results presented in this article, the next step involves employing various methods including the instrumental variable method, lagged variable method, Propensity Score Matching (PSM) method, and a multicollinearity test. These methods are utilized to conduct further research, and a robustness regression is performed based on the model established in the preceding section to further study and test the findings.

#### 4.4.1. Instrumental Variable Method

To address endogeneity concerns, this study employs the two-stage least squares (2SLS) regression method for re-estimation. The annual average equity pledge ratio of the province in which a company is situated (Mean_Pledge) is utilized as the instrumental variable for the equity pledge ratio (Pledge). This index meets two conditions as an instrumental variable: first, the provincial average share pledge ratio is correlated with the share pledge ratio of each local enterprise and has a positive relationship; second, in the case of controlling relevant variables at the enterprise and industry levels, it has no direct impact on the performance of enterprise ESG responsibility. In addition, the instrumental variable passed the endogeneity, under-recognition, and weak recognition tests, which means that the instrumental variable selection is effective and appropriate. The results are presented in Table 6.

Table 6 presents the results of the two-stage regression using instrumental variables. In the first-stage regression, the coefficient (β1) between the average equity pledge ratio of the province (Mean_Pledge) and the equity pledge ratio of the major shareholders of the company (Pledge) is 0.9621, demonstrating a significant positive correlation at the 1% level. This indicates the validity of the instrumental variable. In the second-stage regression, the coefficients (β1) between the equity pledge of major shareholders (Pledge) and corporate

ESG performance (ESG), environmental governance performance (E), social responsibility performance (S), and corporate governance performance (G) are $-0.0227$, $-0.0221$, $-0.0221$, and $-0.0456$, respectively. These coefficients are significant at the 1% level, suggesting that the findings of this study remain robust even after addressing endogeneity concerns.

**Table 6.** Endogeneity and robustness test: instrumental variable method.

| VarName | Step One | Step Two | | | |
| --- | --- | --- | --- | --- | --- |
| | Pledge | ESG | E | S | G |
| Mean_Pledge | 0.9621 *** | −0.0227 *** | −0.0221 *** | −0.0221 *** | −0.0456 *** |
| | (16.0820) | (−2.7964) | (−2.7128) | (−2.7128) | (−4.5863) |
| _cons | 3.2162 | −355.9686 *** | −361.3949 *** | −361.3949 *** | −89.9605 * |
| | (0.9124) | (−8.6332) | (−8.7060) | (−8.7060) | (−1.7492) |
| Control | Yes | Yes | Yes | Yes | Yes |
| year | Yes | Yes | Yes | Yes | Yes |
| Ind | Yes | Yes | Yes | Yes | Yes |
| N | 14,757 | 14,757 | 14,757 | 14,757 | 14,757 |
| R-squared | 0.1159 | 0.1529 | 0.1709 | 0.1968 | 0.2110 |
| Root MSE | 13.525 | 1.0241 | 1.0540 | 1.0335 | 1.3119 |

t statistics in parentheses; * $p < 0.1$ and *** $p < 0.01$.

### 4.4.2. Delayed Primary Variable

Lagged variables can mitigate endogeneity issues to some extent. Hence, this study conducts the regression again with the explanatory variables and all continuous control variables lagged by one period. The results are presented in Table 7.

**Table 7.** Endogeneity and robustness tests: one-phase lag of variables.

| VarName | (1) | (2) | (3) | (4) |
| --- | --- | --- | --- | --- |
| | ESG | E | S | G |
| L.Pledge | −0.0184 *** | −0.0016 | −0.0178 *** | −0.0332 *** |
| | (−15.2615) | (−1.1368) | (−14.7699) | (−24.3522) |
| L.Size | 0.2326 *** | 0.2655 *** | 0.2273 *** | 0.1537 *** |
| | (11.4341) | (10.4933) | (11.1751) | (6.8667) |
| L.Age | 47.0044 *** | 22.0625 *** | 48.0759 *** | 7.9220 |
| | (7.5865) | (3.0393) | (7.7205) | (1.0651) |
| L.Cfo | 2.0129 *** | 0.9859 *** | 1.9249 *** | 2.5854 *** |
| | (9.0563) | (4.0117) | (8.6086) | (9.5073) |
| L.Lev | −1.0591 *** | 0.0523 | −0.9257 *** | −1.7529 *** |
| | (−9.9619) | (0.4145) | (−8.5832) | (−13.9736) |
| L.Growth | −0.0008 | −0.1479 *** | −0.0102 | 0.0838 * |
| | (−0.0192) | (−3.1607) | (−0.2437) | (1.6587) |
| L.Top1 | 0.0037 *** | −0.0025 | 0.0034 ** | 0.0100 *** |
| | (2.6286) | (−1.5707) | (2.3848) | (6.4895) |
| L.Rind | 1.0848 *** | 0.1605 | 0.9469 *** | 2.0576 *** |
| | (4.9980) | (0.6377) | (4.3588) | (8.5461) |
| L.Board | −0.0102 | 0.0074 | −0.0079 | −0.0225 *** |
| | (−1.6338) | (0.9542) | (−1.2505) | (−3.0849) |
| _cons | 1.2433 *** | −0.3207 | 1.3764 *** | 3.7117 *** |
| | (4.6181) | (−1.1113) | (5.1470) | (7.9246) |
| Ind | Yes | Yes | Yes | Yes |
| year | Yes | Yes | Yes | Yes |
| N | 9851 | 9851 | 9851 | 9851 |
| adj. R2 | 0.180 | 0.161 | 0.229 | 0.252 |

t statistics in parentheses; * $p < 0.1$, ** $p < 0.05$, and *** $p < 0.01$.

As shown in Table 7, when the equity pledge ratio is lagged by one period (L.Pledge), the correlation coefficients ($\beta 1$) between corporate ESG performance (ESG), social responsi-

bility performance (S), and corporate governance performance (G) and the equity pledge ratio of major shareholders (Pledge) are −0.0184, −0.0178, and −0.0332, respectively, all significantly negative at the 1% level. However, the correlation coefficient (β1) between environmental governance performance (E) and the equity pledge ratio of major shareholders (L.Pledge) is −0.0016, and the relationship between the two is not significant. This indicates that the regression coefficient (β1) remains significantly negative at the 1% level, affirming that the conclusion holds even after considering endogeneity problems.

### 4.4.3. Propensity Score Matching (PSM)

The Propensity Score Matching (PSM) method is employed to match samples of major shareholders' equity pledges based on propensity scores, dividing them into high-pledge groups and low-pledge groups according to the degree of equity pledged. The equity pledges of major shareholders are categorized into experimental and control groups: the experimental group represents controlling shareholders facing a higher degree of pledge, while the control group comprises controlling shareholders facing a lower degree of pledge. A 1:1 match is conducted to ensure balance between the two groups, followed by the re-regression of the paired sub-samples in the model (1). The regression results are presented in Table 8.

**Table 8.** Endogeneity and robustness tests: Propensity Score Matching (PSM).

| VarName | (1) | (2) | (3) | (4) |
|---|---|---|---|---|
| | ESG | E | S | G |
| Pledge | −0.0189 *** | −0.0011 | −0.0189 *** | −0.0330 *** |
| | (−15.8755) | (−0.7784) | (−15.8755) | (−24.4794) |
| Size | 0.3359 *** | 0.2571 *** | 0.3359 *** | 0.2309 *** |
| | (16.5434) | (10.0766) | (16.5434) | (9.5776) |
| Age | 63.7890 *** | 24.4938 *** | 63.7890 *** | 16.0920 ** |
| | (9.9994) | (3.2923) | (9.9994) | (2.1216) |
| Cfo | 1.5523 *** | 0.5740 ** | 1.5523 *** | 2.3049 *** |
| | (7.3169) | (2.5392) | (7.3169) | (9.4588) |
| Lev | −1.3687 *** | 0.0636 | −1.3687 *** | −2.4686 *** |
| | (−13.0564) | (0.5451) | (−13.0564) | (−20.1428) |
| Growth | −0.1546 *** | −0.1353 *** | −0.1546 *** | −0.1005 * |
| | (−3.6976) | (−3.1081) | (−3.6976) | (−1.9385) |
| Top1 | 0.0047 *** | −0.0025 | 0.0047 *** | 0.0121 *** |
| | (3.2565) | (−1.5647) | (3.2565) | (7.7608) |
| Rind | 0.9036 *** | −0.1961 | 0.9036 *** | 2.1354 *** |
| | (4.2010) | (−0.7784) | (4.2010) | (8.7997) |
| Board | −0.0134 ** | 0.0087 | −0.0134 ** | −0.0321 *** |
| | (−2.1045) | (1.2343) | (−2.1045) | (−4.2738) |
| _cons | −484.4179 *** | −186.2314 *** | −484.4179 *** | −118.6322 ** |
| | (−9.9752) | (−3.2884) | (−9.9752) | (−2.0531) |
| Ind | Yes | Yes | Yes | Yes |
| year | Yes | Yes | Yes | Yes |
| N | 10,309 | 10,309 | 10,309 | 10,309 |
| adj. R2 | 0.217 | 0.172 | 0.217 | 0.298 |

t statistics in parentheses; * $p < 0.1$, ** $p < 0.05$, and *** $p < 0.01$.

Table 8 demonstrates that within the matched samples, the equity pledge of controlling shareholders continues to exhibit a significant negative correlation with the enterprise's ESG performance. Further scrutiny reveals that the equity pledge of controlling shareholders diminishes the company's social responsibility performance (S) and governance level (G), while the association between the equity pledge of controlling shareholders and environmental governance performance (E) remains inconclusive. This consistency with the earlier findings indicates the stability of the regression results. Thus, the conclusion of this paper remains robust even after accounting for endogeneity issues.

4.4.4. Multicollinearity Test

A regression analysis accounts for the issue of multicollinearity, which arises from high correlations between independent variables, resulting in unstable coefficient estimations and unreliable hypothesis testing in the regression model. Multicollinearity can be diagnosed by calculating the Variance Inflation Factor (VIF), as presented in Table 9.

**Table 9.** Endogeneity and robustness tests: multicollinearity tests.

| Variable | VIF | 1/VIF |
|----------|-----|-------|
| Size | 1.46 | 0.685584 |
| Lev | 1.43 | 0.699525 |
| Board | 1.13 | 0.882931 |
| Cfo | 1.08 | 0.925494 |
| Rind | 1.07 | 0.931595 |
| Age | 1.07 | 0.934911 |
| Top1 | 1.05 | 0.952354 |
| Pledge | 1.03 | 0.970663 |
| Growth | 1.02 | 0.984522 |
| Mean VIF | 1.15 | |

The VIF value is a measure of the severity of multicollinearity. It is generally believed that if the VIF value is greater than 10, there is a multicollinearity problem (strictly greater than 5).

The variance inflation factor of the i-th regression coefficient can be expressed as

$$\text{VIF}_i = 1/1 - R_i^2 \tag{2}$$

where $R_i^2$ represents the determination coefficient obtained by fitting the regression equation with the i-th variable as the dependent variable and the remaining independent variables. The larger the VIF value, the stronger the correlation between this variable and the remaining independent variables.

From the analysis results in Table 9, it can be seen that the VIF values of all variables are less than 10, so there is no multicollinearity.

*4.5. Additional Analysis*

4.5.1. Action Path Analysis

If the degree of equity pledged by controlling shareholders continues to increase, it will entail a higher risk of control transfer. Consequently, the controlling shareholder desires the company's stock price to steadily rise as fluctuations in the stock price are closely linked to their operational income. Hence, the controlling shareholder is motivated to engage in earnings management, aiming to increase profits through such practices and thereby enhance the company's operational performance. However, heightened earnings management leads to less reliable earnings disclosures in the company's financial reports. External investors, lacking complete information, are misled by the manipulated earnings data, mistakenly perceiving the company's operational status as favorable and making uninformed investments, ultimately harming investor interests. This form of earnings manipulation demonstrates irresponsibility towards investors and society, adversely affecting the company's social responsibility performance and consequently diminishing the enterprise's overall ESG performance. The following model will be constructed for further analysis:

$$\text{ESG}_{i,t} = \alpha_0 + \alpha_1 \text{Pledge}_{i,t} + \alpha \text{Control}_{i,t} + \sum \text{Year} + \sum \text{Ind} + \varepsilon_{i,t} \tag{3}$$

$$\text{AEM}_{i,t} = \beta_0 + \beta_1 \text{Pledge}_{i,t} + \beta \text{control}_{i,t} + \sum \text{Year} + \sum \text{Ind} + \varepsilon_{i,t} \tag{4}$$

$$\text{ESG}_{i,t} = \gamma_0 + \gamma_1 \text{Pledge}_{i,t} + \gamma_2 \text{AEM} + \gamma \text{Control}_{i,t} + \sum \text{Year} + \sum \text{Ind} + \varepsilon_{i,t} \tag{5}$$

The specific test steps are as follows:

(1) Utilizing model (3) to examine the effect of the equity pledge by controlling shareholders on an enterprise's ESG performance. This impact has already been assessed in the preceding analysis through model (1).
(2) Employing model (4) to investigate whether the equity pledge by controlling shareholders influences the earnings management practices of the enterprise.
(3) Utilizing model (5) to assess the combined impact of the equity pledge by controlling shareholders and the intermediary variable (earnings management) on the enterprise's ESG performance.

Table 10 shows the regression results for the pledge of controlling shareholders' equity, earnings management, and ESG performance of the enterprise.

**Table 10.** Equity pledging of major shareholders, earnings management, and corporate ESG performance.

| VarName | ESG | AEM | ESG | E | AEM | E | S | AEM | S | G | AEM | G |
|---|---|---|---|---|---|---|---|---|---|---|---|---|
| Pledge | −0.0167 *** (−16.1610) | −0.0013 *** (−3.9894) | −0.0165 *** (−16.0223) | −0.0006 (−0.5240) | −0.0013 *** (−3.9894) | −0.0006 (−0.5140) | −0.0162 *** (−15.7420) | −0.0013 *** (−3.9894) | −0.0161 *** (−15.5988) | −0.0314 *** (−27.2320) | −0.0013 *** (−3.9894) | −0.0312 *** (−27.2089) |
| AEM | | | 0.1483 *** (7.8000) | | | 0.0095 (0.4733) | | | 0.1519 *** (7.9471) | | | 0.1957 *** (8.4346) |
| _cons | 1.2699 *** (4.4172) | 0.3293 *** (3.8264) | 1.2210 *** (4.3258) | −0.0305 (−0.1268) | 0.3293 *** (3.8264) | −0.0337 (−0.1396) | 1.3392 *** (4.6503) | 0.3293 *** (3.8264) | 1.2892 *** (4.5493) | 3.9082 *** (8.3042) | 0.3293 *** (3.8264) | 3.8437 *** (8.3685) |
| Control | Yes | Yes | Yes | Yes | Yes | Yes | Yes | Yes | Yes | Yes | Yes | Yes |
| year | Yes | Yes | Yes | Yes | Yes | Yes | Yes | Yes | Yes | Yes | Yes | Yes |
| Ind | Yes | Yes | Yes | Yes | Yes | Yes | Yes | Yes | Yes | Yes | Yes | Yes |
| N | 14,757 | 14,757 | 14,757 | 14,757 | 14,757 | 14,757 | 14,757 | 14,757 | 14,757 | 14,757 | 14,757 | 14,757 |
| Sobel Test | $z = -4.809, p = 0.000$ | | | $z = -1.113, p = 0.266$ | | | $z = -4.787, p = 0.000$ | | | $z = -4.745, p = 0.000$ | | |
| Bootstrap Test | ind_eff:z = −4.79 dir_eff:z = −28.67 | | | ind_eff:z = −1.13 dir_eff:z = 0.18 | | | ind_eff:z = −4.73 dir_eff:z = −30.25 | | | ind_eff:z = −4.78 dir_eff:z = −43.08 | | |

t statistics in parentheses; *** $p < 0.01$.

The test results regarding the mediating effect of earnings management are summarized in Table 10. An analysis of the table data reveals the following:

Model (3) test outcomes align with the original regression results, consistently demonstrating that equity pledges by controlling shareholders significantly hamper the enterprise's ESG performance.

Model (4) testing indicates a significantly negative regression coefficient of equity pledging by controlling shareholders on earnings management (AEM) at the 1% significance level, affirming that this equity pledge enhances the enterprise's earnings management practices.

Model (5) testing reveals that the regression coefficients for earnings management (AEM) are all significantly positive at the 1% significance level, indicating that earnings management has a positive impact on the enterprise's ESG performance.

Additionally, the regression coefficient of the equity pledged by controlling shareholders (Pledge) is significantly negative in Model (5), with its absolute value being less than the absolute value of the regression coefficient in Model (1), suggesting the establishment of a mediating effect.

A further analysis using the Sobel Test and Bootstrap Test demonstrates that the z-value of the mediating effect test is significant at the 1% level, confirming that earnings management indeed serves as a mediating factor in this context.

4.5.2. Analysis of the Heterogeneity of Property Rights

The growth trajectories of listed companies are influenced not only by their internal governance mechanisms but are also shaped by the external environments in which they operate. Private enterprises face heightened risks of control transfer and encounter greater financing challenges following the pledging of controlling shareholders' equity compared to state-owned enterprises. This dynamic often motivates them to undertake a series of strategic decisions influencing the environmental, social, and governance (ESG) performance of the enterprise. To assess whether the impact of controlling shareholders' equity pledging on enterprise ESG performance varies based on the heterogeneity of property rights, this study introduces a dummy variable, Soe, in addition to the variables used in

Model (1). The Soe variable takes a value of 1 for state-owned enterprises and 0 for private enterprises. After incorporating this dummy variable, the model is formulated as follows:

$$\text{ESG}_{i,t} = \alpha_0 + \alpha_1 \text{Soe}_{i,t} + \alpha \text{control}_{i,t} + \sum \text{Year} + \sum \text{Ind} + \varepsilon_{i,t} \qquad (6)$$

To further explore the influence of property rights on the pledging of controlling shareholders' equity, this study segregates the sample into state-owned and non-state-owned categories for a regression analysis. Given the extensive regression data, the results for the state-owned enterprise group and the non-state-owned enterprise group are reported separately for clarity. The regression outcomes for the state-owned enterprise sample are presented in Table 11, while those for the non-state-owned enterprise sample are detailed in Table 12.

**Table 11.** Empirical results for the major shareholder equity pledges and ESG performance of enterprises: a sample of state-owned enterprises.

| VarName | (1) ESG | (2) E | (3) S | (4) G |
|---|---|---|---|---|
| Pledge | −0.0181 ** | −0.0007 | −0.0170 ** | −0.0299 *** |
| | (−5.8870) | (−0.2318) | (−5.4981) | (−8.3244) |
| Size | 0.3079 *** | 0.2950 *** | 0.3052 *** | 0.2572 *** |
| | (9.1583) | (7.2422) | (8.9330) | (6.6776) |
| Age | 16.3803 *** | 8.4715 | 16.6518 *** | −10.0288 *** |
| | (1.6282) | (0.6734) | (1.6349) | (−0.8511) |
| Cfo | 1.1923 *** | 0.4687 | 1.1974 *** | 1.5569 *** |
| | (3.6666) | (1.2355) | (3.5900) | (3.7936) |
| Lev | −1.4191 *** | −0.3413 | −1.3479 *** | −2.1915 *** |
| | (−7.7246) | (−1.5187) | (−7.1856) | (−10.5638) |
| Growth | −0.1764 *** | −0.1320 * | −0.1471 ** | −0.0739 |
| | (−2.6930) | (−1.8233) | (−2.2108) | (−0.8912) |
| Top1 | 0.0012 | −0.0020 | 0.0010 | 0.0075 *** |
| | (0.5435) | (−0.7462) | (0.4370) | (2.8609) |
| Rind | 0.9045 ** | 0.1952 | 0.9038 ** | 1.8708 *** |
| | (2.5646) | (0.4229) | (2.4877) | (4.6166) |
| Board | −0.0041 | −0.0003 | −0.0025 | −0.0087 |
| | (−0.4107) | (−0.0254) | (−0.2494) | (−0.7521) |
| _cons | 1.1102 ** | −0.5456 | 1.2291 ** | 3.9163 *** |
| | (2.4999) | (−1.3983) | (2.2296) | (7.4617) |
| Ind | Yes | Yes | Yes | Yes |
| year | Yes | Yes | Yes | Yes |
| N | 3239 | 3239 | 3239 | 3239 |
| adj. R2 | 0.214 | 0.190 | 0.232 | 0.267 |

t statistics in parentheses; * $p < 0.1$, ** $p < 0.05$, and *** $p < 0.01$.

Tables 11 and 12 present specific regression results concerning the pledging of controlling shareholders' equity and the ESG performance of the state-owned enterprise sample and the non-state-owned enterprise sample, respectively. Examining Table 11, it is evident that in the state-owned enterprise sample, the regression coefficients (β1) of the pledge of the controlling shareholders' equity (Pledge) with the ESG performance of the enterprises (ESG), environmental governance performance (E), and social responsibility performance (S) are −0.0181 and −0.0170, respectively, which are significantly negative at the 5% level. Additionally, the regression coefficient (β1) of the pledge of controlling shareholders' equity (Pledge) with social responsibility performance (S) is −0.0299, which is significantly negative at the 1% level. However, the regression coefficient (β1) of the pledge of controlling shareholders' equity (Pledge) with environmental governance performance (E) did not pass the significance test. Thus, in the state-owned enterprise sample, the pledge of controlling shareholders' equity exhibits a less significant correlation with the ESG performance of en-

terprises and their social responsibility performance (S) while only displaying a significant negative relationship with their corporate governance performance (G).

**Table 12.** Empirical results of equity pledge of major shareholders and the ESG performance of enterprises: a sample of non-state-owned enterprises.

| VarName | (1) | (2) | (3) | (4) |
|---|---|---|---|---|
| | ESG | E | S | G |
| Pledge | −0.0168 *** | −0.0010 | −0.0166 *** | −0.0300 *** |
| | (−14.8001) | (−0.7112) | (−14.6324) | (−23.2185) |
| Size | 0.2537 *** | 0.2408 *** | 0.2485 *** | 0.1692 *** |
| | (12.8019) | (9.5880) | (12.5722) | (7.8326) |
| Age | 62.0936 *** | 14.2733 * | 63.5588 *** | 27.6182 *** |
| | (9.3848) | (1.7754) | (9.4947) | (3.6608) |
| Cfo | 1.4824 *** | 0.6436 *** | 1.4046 *** | 2.2666 *** |
| | (7.5665) | (3.0743) | (7.1792) | (9.9900) |
| Lev | −1.2139 *** | 0.1509 | −1.0507 *** | −2.2993 *** |
| | (−11.9885) | (1.3423) | (−10.2330) | (−19.9180) |
| Growth | −0.1345 *** | −0.1164 *** | −0.1376 *** | −0.1012 ** |
| | (−3.4924) | (−2.8079) | (−3.5495) | (−2.1725) |
| Top1 | 0.0049 *** | −0.0025 * | 0.0047 *** | 0.0104 *** |
| | (3.6095) | (−1.6748) | (3.4553) | (7.4083) |
| Rind | 1.0290 *** | 0.0690 | 0.9572 *** | 1.9461 *** |
| | (5.2264) | (0.3144) | (4.8455) | (8.5818) |
| Board | −0.0177 *** | 0.0071 | −0.0174 *** | −0.0410 *** |
| | (−2.9301) | (0.9701) | (−2.8503) | (−5.8055) |
| _cons | 1.2642 *** | 0.1026 | 1.3477 *** | 4.1527 *** |
| | (3.7863) | (0.3479) | (4.0626) | (7.5969) |
| Ind | Yes | Yes | Yes | Yes |
| year | Yes | Yes | Yes | Yes |
| N | 11,518 | 11,518 | 11,518 | 11,518 |
| adj. R2 | 0.191 | 0.165 | 0.237 | 0.300 |

t statistics in parentheses; * $p < 0.1$, ** $p < 0.05$, and *** $p < 0.01$.

Analyzing Table 12, it is evident that in the non-state-owned enterprise sample, the regression coefficients (β1) of the pledge of controlling shareholders' equity (Pledge) with the ESG performance of the enterprises (ESG), social responsibility performance (S), and corporate governance performance (G) are −0.0168, −0.0166, and −0.0300, respectively. These coefficients are significantly negative at the 1% level. Thus, in the non-state-owned enterprise sample, the pledge of controlling shareholders' equity significantly diminishes the ESG performance of the enterprise. Overall, the impact of the pledge of controlling shareholders' equity on the ESG performance of the enterprise is notably more pronounced in the non-state-owned enterprise sample, suggesting a greater adverse effect on non-state-owned enterprises.

## 5. Discussion and Implications

Under the circumstance of major shareholders pledging equity, the ESG performance of enterprises decreases significantly. Therefore, enterprises should optimize their ownership structure according to their actual conditions, keep the ownership concentration at an appropriate and reasonable level, pay attention to mutual restraint and restraint between authorities, establish an effective equity balance mechanism, and avoid the phenomenon of shareholder dominance so as to constrain controlling shareholders, strengthen the supervision of their behavior, and restrain their equity-pledging behavior for private interests. Governance structures should be improved to strengthen ESG performance.

According to this study, we recognize that enterprises should attach importance to the governance effect of internal control, create a good internal environment, take necessary measures to restrict the equity-pledging activities of major shareholders, enhance the

transparency of corporate information disclosure, curb the equity-pledging behavior of shareholders for private interests, and improve the financing dilemma caused by the deterioration of information asymmetry caused by equity pledging so as to ensure that enterprises have more sufficient cash streams to assume ESG responsibilities.

## 6. Conclusions and Recommendations

This article employs multiple linear regression methods to investigate the influence of major shareholder equity pledges on corporate environmental, social, and governance (ESG) performance. It also examines the heterogeneous impact of property rights and the intermediary role of earnings management. The research findings are as follows: Major shareholder equity pledges mainly reduce a company's social responsibility and governance performance, and major shareholder equity pledges are negatively related to corporate ESG performance. Under the same conditions, compared with state-owned enterprises, the equity pledge behavior of the major shareholders of private enterprises has a more significant impact on corporate ESG. The relationship between earnings management, major shareholder equity pledges, and corporate ESG performance plays an important role. Equity pledges by major shareholders will increase the degree of earnings management and reduce corporate ESG performance.

However, this study has some limitations. This study only takes the perspective of the Chinese market, and determining how to extrapolate these results to other geoeconomic regions around the world is our task to complete. For example, equity pledging by major shareholders has a negative impact on ESG performance. How do these findings apply to other geographic regions? How can this research approach be applied to other geographic regions? These issues are the direction of our future efforts.

Based on the above conclusions, the following suggestions are put forward:

First, major shareholders should realize that an equity pledge is a "double-edged sword" and must carry out reasonable pledges instead of blindly financing through pledges. In addition, major shareholders should insist on self-supervision, avoid behaviors that harm the interests of the company, and maintain the benign operation of the company rather than causing the company to deviate from its normal operational track in order to relieve their own pledge pressure. Only in this way can the company's stock price remain stable and the transfer of control rights be avoided.

Second, the company should establish the concept of ESG performance and realize that ESG construction is needed for its own sustainable development, especially the construction of the environment, to focus on the long-term development of the company.

Finally, listed companies should pay attention to the construction of their external communication platforms and formulate a perfect ESG strategy. Enterprises should establish an ESG control structure according to their own situation, thoroughly evaluate their present situation, and determine the long-term and short-term goals of their ESG strategy through cooperation with internal stakeholders so as to promote the sustainable development of the enterprise.

**Author Contributions:** Methodology, K.Y.; Software, T.Z.; Validation, C.Y.; Resources, K.Y.; Writing—original draft, T.Z.; Writing—review & editing, T.Z.; Supervision, K.Y. and C.Y. All authors have read and agreed to the published version of the manuscript.

**Funding:** National Social Science Fund Project:Study on the influence of controlling shareholder equity pledge on the financial behavior of listed companies: 19BGL070.

**Institutional Review Board Statement:** Not applicable.

**Informed Consent Statement:** Not applicable.

**Data Availability Statement:** Data are contained within the article.

**Conflicts of Interest:** The authors declare no conflict of interest.

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
