# Peer review of "The Sustainability of Corporate ESG Performance: An Empirical Study"

_sustainability, doi:10.3390/su16062377_

Round 1
Reviewer 1 Report
Comments and Suggestions for Authors
As a reviewer, I have several critiques of the paper titled "The Sustainability of Corporate ESG Performance: An Empirical Study":
While the paper suggests a relationship between major shareholder equity pledges and corporate ESG performance, it falls short of establishing causality. Correlation does not imply causation, and without proper controls and identification strategies, it's challenging to attribute changes in ESG performance solely to major shareholder equity pledges. The paper should address potential endogeneity issues and consider alternative explanations for the observed associations.
The paper lacks a robust theoretical framework that guides the research questions and hypotheses.
The study's contribution is questionable due to it is repeated from many studies that dealt with the same topic, which weakens the importance of the study. for example:
Li, Y., & Zhu, D. (2022). Share pledging and corporate environmental investment. Finance Research Letters, 50, 103348.
Cheng, T. Y., Susan, E. B., Lin, H., & Luo, D. (2024). The Relationship between Share Pledge and Corporate Performance: Does Corporate Governance Matter?. Research in International Business and Finance, 102276.
The presentation of findings lacks clarity and precision. There is no strong theoretical explanation for the results, which weakens their contribution to the literature
Comments on the Quality of English Language
Extensive editing of English language required
Reviewer 2 Report
Comments and Suggestions for Authors
The study analyzes the relationship between major shareholders' equity pledging and corporate ESG performance using a dataset of Chinese companies.
The method explores different econometric specifications. In addition, the authors discuss endogeneity issues. Suggestions for improvement - Include references to support the control variables. - Discuss specific characteristics of the Chinese companies that make the study of ESG in this market more appealing. - Tables should be enhanced. For instance, (i) Table 3 is difficult to read, (ii) What/Why is the value 0.0000 (.) for Age in Table 4 and Table 11. - Are there any effects of the covid-19 pandemic in the results? - The conclusion section should include limitations of the study and highlight the implications for theory and practice of ESG or sustainability. Comments on the Quality of English LanguageThe manuscript will benefit from language editing.
Reviewer 3 Report
Comments and Suggestions for Authors
The authors present a very interesting and well detailed research on The Sustainability of Corporate ESG Performance. The results are coherent with the description across the developed research. Nevertheless, important updates must be made.
1- The abstract must be reformulated. Usually the abstract must have a problem statement, a very brief point od situation, the proposed model/approach/other, and a brief presentation of major findings! Furthermore, the language used in the abstract must be easy to grasp by any reader regardless of education background in order to increase the number of potential readers and clear explaining the major objective of the proposed research. Furthermore the authors must bounder the research to the geographical-economical area where data for the study has been gathered. These details must be mentioned in the abstract. For example that the research was conducted in China only, and used the WING database and further CSMAR, short for China Stock Market & Accounting Research Database and 14,757 observations were used.
2- The first paragraphs of the introduction are not referenced… There are many statements done by the researchers that need support from actual valid references. For example, strong statements such as:
Unlike traditional financial metrics, the ESG framework offers a holistic evaluation of a company's performance, taking into account its environmental, social, and governance practices. Must be referenced!
3- The chapters description is not very properly named. For example, 2. Literature Review, and 2.3. Literature Review …. The question here is: Are the following chapters not part of the literature review?
2.1. Research on the Economic Consequences of Major Shareholders’Equity Pledging 111
2.1.1. The Impact of Major Shareholders’ Equity Pledging on Listed Companies
2.1.2. The influence of major shareholders' equity pledge on stakeholders
2.2. Research on the Influencing Factors of Enterprise ESG Performance
4- Unnecessary details: For example the authors must not explicit what software has been used to perform data managed as it is described in lines 319-321.
5- There are too many tables (12 Tables) across the document, it would be better to place some of them in one Anex. It would easy the reading and get more attention from potential readers!
6- In the conclusions chapter is missing the academic and managerial implications of the study. Also, the further works suggestion is missing…
7- From 46 references, 21 are older than 5 or more years. Regarding the ESG topic (which usually is always very strongly updated almost in a quarterly basis), some of the 21 references should updated.
8- The authors should explain how these interesting results can be extrapolated to other geographical-economic areas worldwide. For example, the authors conclude that Major shareholder equity pledges negatively affect corporate ESG performance. How could these findings be used in other geographical regions? How could this research approach be used in other geographical regions?
Good luck
Reviewer 4 Report
Comments and Suggestions for Authors
Reviewer 5 Report
Comments and Suggestions for Authors
I have carefully reviewed the manuscript titled "The Sustainability of Corporate ESG Performance: An Empirical Study," which embarks on an important journey to unravel the complexities between equity pledging by significant shareholders and its implications on corporate Environmental, Social, and Governance (ESG) performance. This exploration, set against the backdrop of A-share listed companies in China from 2014 to 2022, not only enriches the academic discourse but also offers practical insights for stakeholders navigating the nuanced landscape of corporate sustainability.
Recommendations:
The manuscript's foundation on the relationship between equity pledging and ESG performance is commendably outlined. Yet, it would greatly benefit from a deeper theoretical anchorage. Drawing connections to foundational theories such as agency theory or stakeholder theory could provide a richer context for understanding the mechanisms at play. This theoretical grounding could further bolster the study's hypothesis development, providing a clearer pathway from theoretical expectations to empirical investigation.
The methodological rigor evident in the selection of databases and the analytical approach is laudable. However, the manuscript would be enhanced by a more detailed justification for the chosen timeframe and its relevance to the study's objectives. Additionally, the rationale behind excluding certain industries and the specific criteria for data cleansing and outlier management (e.g., Winsorization) demand further elucidation. This would not only increase transparency but also reinforce the credibility of the study's findings.
The empirical analysis is undeniably thorough. Nevertheless, addressing potential endogeneity concerns beyond the methods already employed (e.g., instrumental variable approach, lagged variables) could further solidify the findings. Discussing alternative econometric models or sensitivity analyses that test the robustness of the results under different assumptions would be invaluable. This could include exploring the effects of unobserved heterogeneity or dynamic relationships over time.
This is a great job about ESG Performance. However, there are a couple of newest working papers related to your research that you should consider to enrich your literature review and catch up the forefront of the research in this area:
Garcia, F., Tsvetelin Gankova-Ivanova, T., González-Bueno, J., Oliver, J., Tamošiūnienė, R. 2022. What is the cost of maximizing ESG performance in the portfolio selection strategy? The case of The Dow Jones Index average stocks. Entrepreneurship and Sustainability Issues, 9(4), 178-192.
García, F., González-Bueno, J., Guijarro, F., & Oliver, J. (2020). Forecasting the Environmental, Social, and Governance Rating of Firms by Using Corporate Financial Performance Variables: A Rough Set Approach. Sustainability, 12(8), 3324.
In essence, this manuscript contributes significantly to our understanding of the intricate dynamics between equity pledging and corporate ESG performance. With the above enhancements, I am confident that it will not only provide a robust academic contribution but also offer meaningful guidance for enhancing corporate sustainability practices.
Round 2
Reviewer 1 Report
Comments and Suggestions for Authors
Unfortunately, the paper is still need improvement, and I do not think it meets the standards of academic publishing in this form.
This can be seen from the first line in the abstract.
(environmental, social and government)
government!!!
It is necessary to develop the paper linguistically.....
Also, the contribution of the study needs to be clarified in terms of its importance and what distinguishes it from previous studies, as the authors’ claim that they divided the ESG factors into ESG separately is not enough. Many literatures have done the same work previously, for example:
Al Amosh, H., Khatib, S. F., & Ananzeh, H. (2024). Terrorist attacks and environmental social and governance performance: Evidence from cross‐country panel data. Corporate Social Responsibility and Environmental Management, 31(1), 210-223.
I also suggested developing the theoretical framework for the study, and i suggest looking at some recent studies for that. For example:
https://doi.org/10.1002/bsd2.240
Comments on the Quality of English LanguageExtensive editing of English language required
Reviewer 3 Report
Comments and Suggestions for Authors
great job
Author Response
Dear Reviewer,
We are writing to express our heartfelt gratitude for your careful review of our paper and your valuable feedback. We are delighted to learn that you have no comments or suggestions on our paper, as this provides us with great encouragement and affirmation.
Your comments and suggestions during the review process have been incredibly helpful to us. We have carefully considered your feedback and made corresponding revisions and improvements to the paper. We believe that these enhancements will further enhance the quality of our paper.
Once again, we would like to express our sincerest thanks for your hard work and dedication to our paper. If you have any further suggestions or comments in the future, we would be more than happy to receive and implement them.
Looking forward to your further guidance!
Reviewer 4 Report
Comments and Suggestions for Authors.
Author Response

(The authors gave the same response as above.)

Round 3
Reviewer 1 Report
Comments and Suggestions for Authors
The authors have done such a good job that the paper can be accepted in its current form